# Chromoanagenesis in Osteosarcoma

**DOI:** 10.3390/biom15060833

**Published:** 2025-06-07

**Authors:** Guozhuang Li, Nan Wu, Jen Ghabrial, Victoria Stinnett, Melanie Klausner, Laura Morsberger, Patty Long, Ezra Baraban, John M. Gross, Ying S. Zou

**Affiliations:** 1State Key Laboratory of Complex Severe and Rare Diseases, Department of Orthopedic Surgery, Peking Union Medical College Hospital, Peking Union Medical College and Chinese Academy of Medical Sciences, Beijing 100730, China; liguozhuang003@gmail.com; 2Beijing Key Laboratory of Big Data Innovation and Application for Skeletal Health Medical Care, Beijing 100730, China; 3Key Laboratory of Big Data for Spinal Deformities, Peking Union Medical College and Chinese Academy of Medical Sciences, Beijing 100730, China; 4Department of Pathology, Johns Hopkins University School of Medicine, Baltimore, MD 21205, USA; jghabri1@jh.edu (J.G.); vlloyd3@jhmi.edu (V.S.); mhardy22@jhmi.edu (M.K.); lmorsber@jhmi.edu (L.M.); plong@jhmi.edu (P.L.); ebaraba1@jhmi.edu (E.B.); jgross28@jhmi.edu (J.M.G.)

**Keywords:** chromoanagenesis, osteosarcoma, structural variant, chromothripsis, chromoanasynthesis, copy number variant, gene mutation

## Abstract

Chromoanagenesis is a catastrophic genomic phenomenon involving sudden, extensive rearrangements within one or a few cell cycles. In osteosarcoma, the most prevalent malignant bone tumor in children and adolescents, these events dramatically alter the genomic landscape, frequently disrupting key tumor suppressor genes like *TP53* and *RB1*, amplifying oncogene expression, and propelling tumor progression and evolution. This review elucidates how key chromoanagenic mechanisms, such as chromothripsis and chromoanasynthesis, arise from replication stress and impaired DNA repair pathways, ultimately contributing to genomic instability in osteosarcoma. Chromothripsis features prominently in osteosarcoma, occurring in up to 62% of tumor regions and driving intratumoral heterogeneity through persistent genomic crises. Next-generation sequencing, optical genome mapping, and emerging technologies like single-cell sequencing empower researchers to detect and characterize these complex structural variants, demonstrating how a single catastrophic event can profoundly influence osteosarcoma progression over time. While targeted therapies for osteosarcoma have proven elusive, innovative strategies harnessing comprehensive genomic profiling and patient-derived preclinical models hold promise for uncovering tumor-specific vulnerabilities tied to chromoanagenesis. Ultimately, unraveling how these rapid, large-scale rearrangements fuel osteosarcoma’s aggressive nature will not only refine disease classification and prognosis but also pave the way for novel therapeutic approaches to enhance patient outcomes.

## 1. Introduction

Osteosarcoma, the most common primary malignant bone tumor, primarily affects children and adolescents [1]. It accounts for approximately 20% of all primary bone malignancies and typically arises during periods of rapid bone growth, most often affecting individuals aged 10 to 20 years [2]. A second incidence peak occurs in older adults, frequently associated with predisposing conditions or prior radiation exposure. Clinically, osteosarcoma manifests as localized bone pain and swelling, with the knee or proximal humerus being the most common sites [3]. In some instances, a pathological fracture serves as the initial indicator of disease. Osteosarcoma often follows a highly aggressive disease course, with a strong tendency for metastatic spread, primarily to the lungs [4]. Diagnosis is established through biopsy, which reveals malignant cells producing an osteoid matrix. Histologically, osteosarcomas are classified into low, intermediate, or high grades [5] (Figure 1). Low-grade tumors, characterized by slower progression, are managed with surgery alone, while high-grade tumors, with their elevated metastatic potential, necessitate a combination of surgical resection and chemotherapy [6]. The effective management of osteosarcoma demands a coordinated multidisciplinary team, including oncologists, surgeons, pathologists, radiologists, and specialized nurses.

Osteosarcoma’s clinical significance lies in its aggressive nature and high propensity for early metastasis, posing major challenges for patient management and long-term outcomes. Over recent decades, advances in diagnostic imaging, surgery, and adjuvant therapies have improved the five-year survival rate to 60–70% for localized disease, but the prognosis remains poor for patients with metastatic or recurrent disease [1]. A deeper understanding of osteosarcoma’s etiology and molecular mechanisms is crucial for developing novel therapies. The sequential steps driving osteosarcoma development are still poorly elucidated, though evidence suggests a process involving *TP53* loss and a pivotal event sparking widespread chromosomal rearrangements [1]. At the chromosomal level, osteosarcoma exhibits complex karyotypic abnormalities and widespread genomic instability, largely driven by key tumor suppressor mutations. These profound alterations yield an unstable genome, promoting tumor heterogeneity and complicating treatment efforts. Pinpointing the drivers of this genomic complexity remains a key challenge in improving patient outcomes.

One important phenomenon driving these extensive genomic alterations is chromoanagenesis (chromosome regeneration), a catastrophic event wherein one or more chromosomes undergo rapid and extensive rearrangements within a single or a few cell cycles [7]. This process plays a critical role in shaping the genomic landscape of osteosarcoma, further exacerbating its complexity. This single-step genomic crisis can profoundly alter copy numbers, disrupt gene integrity, and generate novel fusion genes, thereby accelerating tumor evolution. This mechanism upends the conventional view of gradual tumor progression, proposing instead that cancer can emerge from sudden, large-scale genomic upheavals rather than from a slow accrual of mutations [8]. Certain chromoanagenesis events align with the punctuated equilibrium model, where a solitary cataclysmic event triggers a cascade of oncogenic alterations, swiftly propelling malignancy. These revelations fundamentally redefine our perspective on cancer evolution. Complex chromosomal rearrangements tied to chromoanagenesis manifest across a spectrum of tumors, as well as congenital and developmental disorders [7,9,10]. However, not all chromoanagenesis events are harmful, as copy number variants (CNVs) and complex structural variants (SVs) also occur in phenotypically normal individuals [11]. In tumors, the massive genomic upheavals spurred by chromoanagenesis amplify heterogeneity, shaping the clinical outcomes by influencing tumor aggressiveness, drug resistance, and prognosis. Mounting evidence underscores chromoanagenesis as a pivotal—and at times decisive—driver of cancer development. Still, the exact molecular triggers and clinical ramifications of chromoanagenesis in osteosarcoma remain elusive, highlighting a critical knowledge gap. Unraveling its biological foundations is essential for sharpening cancer classification, pinpointing innovative therapeutic targets, and enhancing patient care.

Innovations in biotechnologies—such as whole-genome sequencing (WGS), long-read sequencing, RNA sequencing (RNA-seq), methylation analysis and epigenomic profiling, single-cell sequencing, and multi-omics approaches—have markedly improved our capacity to dissect the mutational landscape, structural rearrangements, and clonal evolution across a range of cancers. Given osteosarcoma’s profound genomic intricacy, probing the role of chromoanagenesis could yield vital insights into its pathogenesis and unlock pathways for targeted therapies. In this review, we explore the latest insights into chromoanagenesis, its influence on osteosarcoma biology, and approaches for thoroughly characterizing these events.

## 2. Impact of Chromoanagenesis in Osteosarcoma

### 2.1. Complex Genomic Rearrangements and Genetic Instability

A comprehensive analysis of the genomic changes driving osteosarcoma is essential to enhance our understanding and refining treatment strategies for this tumor. Genome-wide sequencing investigations have highlighted the remarkable complexity of osteosarcoma genomes, indicating that only a small subset of recurring genetic alterations is consistently detectable across diverse osteosarcoma cases [12,13].

We reviewed published osteosarcoma cohort studies with accessible, detailed data, applying the following inclusion criteria: (1) clear identification of chromoanagenesis; (2) availability of data on specific chromosomes impacted by chromoanagenesis and associated gene alterations, sourced from the main text or their supplementary materials. Our study included nine cohort studies [12,14,15,16,17,18,19,20], and the summarized results are shown in Figure 2. In osteosarcoma cases with chromoanagenesis, the five most frequently affected chromosomes are chromosomes 12, 1, 8, 6, and 2 (Figure 2A). Regarding the affected genes, the top five involved in SVs are *TP53*, *RB1*, *ATRX, NF1*, and *SRF2* (Figure 2B); in CNVs, the most commonly affected genes are *CDK4* (gain/amplification), *TP53* (loss), *MDM2* (gain/amplification), *RB1* (loss), and *CDKN2A* (loss) (Figure 2C); and in single-nucleotide variants (SNVs), the top five affected genes are *TP53*, *RB1*, *PTEN, PIK3CA*, and *ATRX* (Figure 2D).

Osteosarcoma usually exhibits a relatively low burden of SNVs, with few recurrent alterations in protein-coding genes beyond well-known tumor suppressors such as *TP53* and *RB1* [6,12,13,14]. However, recent studies have identified additional mutational patterns. For instance, Perry et al. identified recurrent mutations in the PI3K/mTOR (phosphatidylinositol-3-kinase/mammalian target of rapamycin) pathway in osteosarcomas, primarily through whole-exome sequencing (WES) [21]. Similarly, Behjati et al. identified mutations in insulin-like growth factor (IGF) signaling genes in 7% (8/112) of osteosarcoma cases and validated them in another independent cohort [12]. In contrast, osteosarcoma is characterized by extensive and recurrent CNVs and SVs, with a median of several hundred SVs per tumor. These aberrations display marked heterogeneity across osteosarcoma cases [14]. The high prevalence of CNVs and SVs suggests that the primary oncogenic drivers in osteosarcoma may reside within these large-scale genomic rearrangements rather than in point mutations [6,12]. Recent research highlights the frequent biallelic inactivation of *TP53*, high rates of whole-genome doubling (WGD), complex chromothripsis events, and abundant CNVs, all of which vary substantially among patients [6,22,23].

Chromoanagenesis primarily arises through micronucleus formation and chromatin bridges [24]. Micronuclei, formed from lagging chromosomes or acentric fragments, exhibit defective nuclear envelopes, leading to DNA replication errors, damage accumulation, and erroneous reassembly into highly rearranged chromosomes. Chromatin bridges, resulting from dicentric chromosomes via telomere fusions, including breakage–fusion–bridge (BFB) cycles, often rupture during mitosis, causing catastrophic fragmentation and chromothripsis-like rearrangements, frequently accompanied by kataegis. Both mechanisms, driven by genomic instability, replication stress, and mitotic defects, are likely key contributors to the extensive chromosomal rearrangements seen in tumors.

#### 2.1.1. Chromothripsis in Osteosarcoma

Chromothripsis is characterized by massive chromosomal shattering followed by random reassembly of fragmented DNA, resulting in complex segmental copy number alterations [12,25,26]. Osteosarcoma is among the most striking examples of a malignancy driven by chromothripsis, which contributes to profound genome complexity [12,14,25]. In a comprehensive analysis of the largest osteosarcoma cohort to date, Valle-Inclan et al. reported that chromothripsis was evident in 62% of the analyzed tumor regions. Strikingly, in 74% of the cases, at least one region exhibited chromothripsis [14]. These findings underscore chromothripsis as a pervasive and dynamic mutational process that persists throughout tumor development.

Building on the understanding that chromothripsis repeatedly disrupts and reshapes the cancer genome, subsequent studies have delineated specific patterns of catastrophic rearrangements that further illuminate its role in tumorigenesis. Behjati et al. identified a distinct genomic signature termed “chromothripsis amplification”, which arises from a single catastrophic event followed by amplification and BFB cycles [12]. This process can generate multiple oncogenic drivers in both pediatric and adult osteosarcoma. Furthermore, derivative chromosomes formed by chromothripsis are often highly unstable, fueling persistent genomic instability, extensive intratumor heterogeneity, and continued genome evolution, including the formation of extrachromosomal circular DNA (ecDNA) [14]. Further insights from Wu et al. revealed a higher incidence of clustered genomic rearrangements consistent with chromothripsis in younger osteosarcoma patients, a trend also observed in the predominantly pediatric TARGET cohort [15]. These findings highlight the potential importance of age-related genomic instability mechanisms in shaping osteosarcoma pathogenesis. Adding to this complexity, Valle-Inclan et al. described a novel rearrangement process termed “loss–translocation–amplification (LTA) chromothripsis,” which underlies some of the most intricate derivative chromosomes observed in high-grade osteosarcomas (HGOSs) [12,14]. This mechanism intersperses chromothripsis patterns with segmental amplifications across multiple chromosomes. In contrast to canonical translocation–bridge amplification or classical BFB cycles, LTA chromothripsis can achieve biallelic *TP53* inactivation and enable tolerance to whole-genome duplication in roughly half of HGOSs, while simultaneously amplifying multiple oncogenes through multi-generational BFB cycles involving multiple chromosomes [27].

The underlying causes of chromothripsis remain incompletely understood and warrant further investigation. Kovac et al. used WES and microarrays to report genomic alterations in osteosarcoma, suggesting that defects in homology-directed DNA repair may contribute to chromothripsis [23]. Gong et al. reported that Ran GTPase-activating protein 1 (RanGAP1) is frequently reduced or inactivated in human osteosarcoma, strongly correlating with elevated chromothripsis [28]. In a mouse model of rapidly proliferating osteoprogenitors, loss of RanGAP1 precipitated chromothripsis on chromosome 1q, resulting in immediate inactivation of Rb1 and degradation of p53. These defects in essential tumor suppressor pathways severely compromised DNA damage repair mechanisms, thereby accelerating osteosarcoma onset [28].

In parallel, Carlo et al. demonstrated that inactivation of Profilin 1 (*Pfn1*) drives chromothripsis through catastrophic mitotic dysregulation in Pagetic osteosarcomas [29]. Their findings revealed that *Pfn1* deficiency disrupts spindle midzone organization, leading to a 42% increase in multipolar spindles, a 68% prevalence of anaphase bridges, and impaired actin filament recruitment to cleavage furrows. This results in a 39% incidence of cytokinesis failure and subsequent micronucleus formation [28]. These defects precipitate the chromothriptic fragmentation of missegregated DNA within micronuclei, followed by clonal selection of survival-advantageous rearrangements. This mechanism underpins the pervasive copy number alterations (observed in 78% of *Pfn1*-deficient tumors) and oscillating genomic patterns characteristic of chromothripsis, establishing mitotic catastrophe as a pivotal link between cytoskeletal dysfunction and chromosomal instability in osteosarcoma pathogenesis.

#### 2.1.2. Other Types of Chromoanagenesis

##### Chromoanasynthesis in Osteosarcoma

Chromoanasynthesis represents a replication-based mechanism of chromosomal restructuring, distinct from chromothripsis and chromoplexy [7,30]. Chromoanasynthesis could represent a broader spectrum of segmental amplification processes, with tandem segmental duplication as the most fundamental component [30]. This process is characterized by localized gains, including duplications and triplications, alongside deletions and copy-neutral segments [24,31]. Unlike chromothripsis, which arises from DNA fragmentation and reassembly, chromoanasynthesis is driven by defective DNA replication, likely involving mechanisms such as fork stalling and template switching (FoSTeS) and microhomology-mediated break-induced replication (MMBIR). While chromoanasynthesis-related rearrangements are primarily observed in constitutional disorders [10,24], Pires et al. found that 5 of 28 primary osteosarcomas from Brazilian patients exhibited multiple localized amplifications at different copy number levels, a pattern consistent with chromoanasynthesis. Two of these patients, both harboring *TP53* and *RB1* mutations, had chromoanasynthesis-like alterations and an above-average CNA burden (63 and 46, respectively) [4]. While chromoanasynthesis is less frequently reported in osteosarcoma than chromothripsis, its role in tumorigenesis warrants further investigation.

##### Chromoplexy in Osteosarcoma

Chromoplexy involves large-scale, interdependent rearrangements that affect multiple chromosomes in a coordinated fashion, which typically involve chained translocations and deletions with minimal CNVs. They frequently disrupt tumor suppressor genes, thereby contributing to cancer progression. In addition to its prominence in prostate cancer [8], chromoplexy has also been identified in several bone and soft tissue tumors, arising via complex, loop-like rearrangements rather than simple translocations, a feature that correlates with poorer clinical outcomes. These observations underscore the pivotal role of complex genomic rearrangements—chiefly chromoplexy, but also chromothripsis—in driving tumor evolution in bone and soft tissue malignancies [32].

### 2.2. Tumor Heterogeneity and Evolutionary Dynamics

Osteosarcoma arises through a multifaceted process encompassing oncogenic events that initiate malignant transformation, a progressive accumulation of genetic aberrations as osteoblast lineage cells proliferate during bone growth [23], and a permissive microenvironment that sustains tumor expansion. The interplay between osteosarcoma cells and their surrounding niche is pivotal for tumor growth within bone [1]. This microenvironment modulates cellular behavior—promoting proliferation, quiescence, invasion, migration, and drug resistance—thereby contributing to osteosarcoma’s marked intrinsic heterogeneity [33,34]. Notably, alterations in the IGF pathway have also been implicated as an age-independent driver of osteosarcoma in both pediatric and adult patients [12].

Although WES and single-cell WGS have revealed substantial karyotypic heterogeneity in osteosarcoma, these methods often lack the sensitivity to detect subclonal SVs and face challenges in accurately defining subclonal somatic copy number changes from single-biopsy samples [35]. In contrast, high-depth multi-region WGS has emerged as a pivotal approach to uncover the influence of complex genomic rearrangements (CGRs) on tumor onset, intratumor heterogeneity, and clonal evolution [14]. Furthermore, in vitro and in vivo models highlight that complex SV processes—including BFB cycles—promote clonal heterogeneity, oncogene amplification, ecDNA formation, and drug resistance [22,27,36,37].

### 2.3. Prognostic Significance and Clinical Outcomes

Osteosarcoma develops through a combination of oncogenic events that trigger malignant transformation, progressive genetic aberrations during rapid osteoblast proliferation, and a supportive microenvironment essential for cancer cell growth [1]. Emerging evidence indicates that chromoanagenesis is activated early during malignant transformation [6,38]. Chromoanagenesis accelerates the acquisition of key driver mutations and further destabilizes the genome, thereby promoting tumor progression.

A critical factor in this process is the biallelic inactivation of *TP53* function. Biallelic inactivation of *TP53* not only impairs cell cycle regulation and apoptosis but also renders the genome more susceptible to catastrophic events, including chromoanagenesis. The resulting chromosomal complexity enhances tumor heterogeneity, which impacts both disease progression and the tumor’s response to therapy. In addition, recent studies have highlighted the prognostic significance of extensive loss of heterozygosity (LOH) in high-grade osteosarcoma [1,14]. LOH may serve as an indirect marker of underlying genomic catastrophes such as chromoanagenesis. This correlation suggests that LOH could provide valuable prognostic information, aiding in the stratification of patients and the customization of therapeutic approaches. In addition, CD95 expression in osteosarcoma is negatively associated with metastatic potential, and reduced CD95 levels are linked to unfavorable prognosis in osteosarcoma patients [39,40]. Overexpression of HER2 is linked to poorer clinical outcomes, but *HER2* is not amplified in osteosarcoma, and its prognostic value remains controversial [41,42,43]. Alterations like *RB1* loss and *MYC* and *VEGFA* amplification may indicate a high risk of osteosarcoma, but none are yet validated for clinical risk stratification [1].

### 2.4. Therapeutic Strategies and Targeted Interventions

Osteosarcoma treatment has long relied on a chemotherapy regimen—doxorubicin, cisplatin, and methotrexate—originally established nearly 40 years ago [44]. Emerging evidence shows that the doxorubicin component can trigger “genome chaos,” i.e., a transient wave of polyploid giant cell formation and wholesale karyotype re-shuffling that seeds drug-tolerant subclones within weeks [45]. Clarifying whether this stress-induced macro-evolutionary burst underlies the rapid relapse observed in a subset of patients will require the serial whole-genome or karyotype profiling of paired pre-/post-treatment samples. To date, no reliable biomarkers have been identified to guide patient stratification for distinct therapeutic options. Given the extensive heterogeneity among osteosarcomas, no single model system can effectively evaluate the therapeutic potential of specific drugs across all cases, underscoring the need for a genome-informed approach. A comprehensive analysis of copy number alterations and their therapeutic relevance in osteosarcoma provides a blueprint for genome-guided research and highlights the promise of precision medicine trials for this disease [6].

Mitigating chromoanagenesis in osteosarcoma calls for interventions that restore or enhance core tumor suppressors (e.g., *TP53*, *RB1*) and DNA damage response pathways (e.g., *ATM*, *ATR)*. These proteins coordinate cell cycle checkpoints and repair mechanisms that minimize genomic instability, thereby reducing the risk of large-scale chromosomal chaos. Integrating treatments aimed at bolstering these pathways with emerging precision medicine strategies may offer a more comprehensive, genome-informed approach to osteosarcoma therapy. Although alterations in the p53 or Rb pathway are common in osteosarcoma, Franceschini et al. demonstrated that the loss of both p53 and p16^Ink4a^ is crucial for transforming mesenchymal stem cells (MSCs) into osteosarcoma cells [46]. Their study explored the *Cdkn2a/Cdkn2b* gene locus in osteosarcomagenesis, finding that MSCs lacking p15^Ink4b^, p16^Ink4a^, or p19^Arf^ transform faster than wild-type MSCs, suggesting that cell cycle dysregulation is key to osteosarcoma initiation. Notably, this research revealed a therapeutic opportunity: CDK4/CDK6 inhibitors, like palbociclib, effectively target p16^INK4A^ loss in osteosarcoma models. Additionally, about 20–23% of primary osteosarcoma biopsies showed intact Rb protein but defective p16^INK4A^ or CDK4/CDK6 overexpression, indicating that CDK4/CDK6 inhibition could benefit a subset of patients. These findings suggest that targeting the CDK4/CDK6 pathway holds promise as a therapeutic strategy for osteosarcoma.

Particular attention should be paid to chromothripsis, as it plays a crucial role in the pathogenesis of osteosarcoma and is associated with poor clinical outcomes and increased therapeutic resistance. Preventing the development of osteosarcoma through chromothripsis mechanisms requires a deep understanding of the underlying genetic disruptions [47]. Chromothripsis in osteosarcoma is often linked to defects in DNA repair pathways, such as those involving *TP53*, *ATM*, *MRE11*, and *WRN*. These genes play pivotal roles in maintaining genomic stability by regulating double-strand break repair and controlling cell cycle checkpoints. Targeting these pathways, particularly through restoring proper DNA damage response mechanisms or enhancing checkpoint control, could reduce the occurrence of chromothripsis and its associated genomic instability [47]. Moreover, defects in DNA double-strand break repair—particularly within the homologous recombination pathway—have been implicated in driving these large-scale chromosomal rearrangements [48,49]. Building on this mechanistic insight, researchers have turned to synthetic lethality as a strategy to exploit DNA repair deficiencies in osteosarcoma cells. In one such effort, Khalid et al. performed a systematic screening of drug combinations designed to selectively kill tumor cells exhibiting chromothripsis while sparing those with intact repair processes. Their findings revealed a potent synergistic interaction between the HDAC inhibitor romidepsin and PARP inhibitors [50]. This combination therapy markedly suppressed tumor growth and triggered apoptosis in patient-derived xenograft models, highlighting the therapeutic potential of co-targeting epigenetic regulation and DNA repair pathways to overcome the aggressive biology of chromothripsis-driven osteosarcoma [50]. Furthermore, preventing telomere crisis and stabilizing chromosomal integrity might mitigate the chromosomal shattering observed in osteosarcoma cells. By addressing the root causes of chromothripsis, therapeutic strategies can be developed to hinder the initiation and progression of osteosarcoma, ultimately improving patient outcomes.

Personalized strategies targeting somatic alterations in osteosarcoma have gained traction as small-molecule inhibitors undergo development and testing in various cancers [51]. Recent regulatory changes by the FDA now allow many of these trials to include patients aged 12 years and older, encompassing most individuals with osteosarcoma. While these histology-agnostic trials facilitate broader patient participation, the diverse spectrum of actionable mutations in osteosarcoma and the limited number of enrolled patients per trial may constrain new insights into the disease’s biology. Additionally, because these are novel agents, data are tightly controlled by trial sponsors, delaying their broader access and limiting immediate advancements in disease understanding. Once these findings become publicly available, improved bioinformatic tools to compile and analyze the data could enhance our knowledge of targeted therapy’s role in osteosarcoma [1].

In sum, we outlined the multifaceted contribution of chromoanagenesis to osteosarcoma initiation, tumor evolution, drug resistance, and potential therapeutic interventions (Figure 3).

## 3. Molecular Basis and Detection of Chromoanagenesis

### 3.1. Molecular Basis of Chromoanagenesis

Chromoanagenesis is primarily driven by DNA damage, particularly double-strand breaks (DSBs), replication stress, and defects in DNA repair pathways. Under normal conditions, cells utilize precise repair mechanisms such as non-homologous end joining (NHEJ) and homologous recombination (HR) to resolve DSBs. However, when these pathways are impaired or overwhelmed, extensive chromosomal fragmentation, illegitimate re-ligation, and complex rearrangements may occur, giving rise to chromoanagenesis.

Replication stress, often induced by oncogene activation or other cellular insults, exacerbates genomic instability by leading to replication fork collapse, additional DSBs, and increased reliance on error-prone repair mechanisms. Notably, the tumor suppressor p53 plays a pivotal role in detecting DNA damage and initiating either cell cycle arrest or apoptosis. Loss of p53 function—a common occurrence in osteosarcoma—enables cells with profound chromosomal abnormalities to evade apoptotic elimination, fostering genomic chaos. Additionally, defects in DNA damage response factors such as *ATM*, *ATR*, and *BRCA1/2* can further promote the accumulation of SVs.

Three primary mechanisms illustrate distinct yet overlapping pathways to chromoanagenesis (Figure 4): chromothripsis (chromosome shattering), chromoplexy (chromosome braiding), and chromoanasynthesis (chromosome reconstitution or reassortment). Each of these phenomena contributes to extensive genomic remodeling, frequently disrupting tumor suppressor genes and amplifying oncogenes, thereby influencing cancer initiation and progression. These chaotic events result in highly remodeled derivative chromosomes. Although each presents distinct molecular hallmarks, they share the common feature of occurring within a single cell cycle and are fundamentally linked to genomic instability [24].

Genome architecture theory (GAT) frames chromoanagenesis as the macro-evolutionary “genome-chaos” phase that punctuates cancer evolution, setting the stage for subsequent stepwise clonal refinement [52,53]. Catastrophic bursts—including chromothripsis, chromoplexy, chromoanasynthesis, PGCC-driven giant nuclei, and large-scale fragmentation—crack open karyotype coding in a single cell cycle, reshuffling gene order and spawning wholly new genomic architectures [54]. The resulting clone pool fuels the rapid emergence of aggressive, treatment-resistant osteosarcoma subclones, making karyotype-level surveillance as critical as tracking point mutations [14].

Although chromoanagenesis arises from stochastic events such as chromosome shattering or replication errors, the survival of the affected cells is not purely random. Only rearrangements that preserve essential genomic functions can sustain cell viability. As Holland and Cleveland emphasize, most deleterious configurations are eliminated, while rare, functionally tolerable architectures persist. This selection is evident in the preservation of heterozygosity in key regions and the spatial confinement of damage within micronuclei [7]. Thus, chromoanagenesis reflects a critical paradox: while the process is chaotic, the outcome is filtered by the requirement for genomic functionality.

### 3.2. Traditional, Advanced, Emergering Technologies to Study Chromoanagenesis

Elucidating the complex rearrangements that define chromoanagenesis requires a broad spectrum of genomic and cytogenetic tools. These can be grouped into three major categories, each offering distinct advantages in detecting and characterizing large-scale chromosomal alterations (Figure 5).

#### 3.2.1. Traditional Cytogenetic Methods

Karyotyping provides an overview of gross chromosomal anomalies, although its resolution is limited [55,56,57]. Fluorescence in situ hybridization (FISH) targets specific genomic loci in metaphase or interphase cells, which makes it useful for confirming suspected rearrangements and pinpointing critical breakpoints [12,58,59]. Spectral karyotyping (SKY) and multicolor FISH (M-FISH), as advanced extensions of traditional FISH, enable full-chromosome painting and the simultaneous visualization of all chromosomes in different colors, allowing for the detection of highly complex, nonclonal structural aberrations associated with genome chaos [60]. Optical genome mapping (OGM) offers higher resolution than conventional karyotyping and can detect complex SVs by visualizing ultra-high-molecular-weight DNA molecules [61]. Meanwhile, Hi-C, a chromosome conformation capture technique, reveals how chromosomes are packaged and can uncover hidden rearrangements by mapping three-dimensional genome architecture [37,62,63].

#### 3.2.2. Array-Based and Sequencing Approaches

Microarray comparative genomic hybridization and SNP arrays are well established for identifying CNVs across the genome [55,59]. However, these platforms may struggle with complex or cryptic rearrangements that span multiple breakpoints. Next-generation sequencing (NGS) tools overcome many of these limitations: short-read genome sequencing (Sr-GS) captures small-scale variations but can be challenged by highly rearranged regions, whereas long-read genome sequencing (Lr-GS) offers a clearer view of extensive or repetitive rearrangements [7,20,64,65,66]. Together, these methods can pinpoint deletions, duplications, and more intricate genomic rearrangements characteristic of chromoanagenesis.

#### 3.2.3. Single-Cell Sequencing Coupled with Multi-Omics Profiling

Although bulk sequencing provides an aggregate snapshot of tumor genomes, single-cell sequencing coupled with multi-omics profiling offers critical insights into intratumoral heterogeneity and the evolutionary trajectory of chromoanagenesis events [67,68,69,70,71,72]. By analyzing the multi-omics content of individual cells, researchers can trace how catastrophic rearrangements arise and expand within distinct subclones, refining our understanding of the selective pressures that drive tumor progression. New platforms (e.g., 10× Multiome, CITE-seq) simultaneously capture genome, transcriptome, and chromatin signals from the same cell, directly linking catastrophic structural variants to their functional outputs in osteosarcoma [73,74]. Long-read single-cell methods (nanoNOMe-seq, scNano-SV) resolve complex breakpoint junctions and local methylation at base-pair resolution [75,76]. Time-resolved CRISPR lineage barcoding indicates that chromoanagenesis can precede whole-genome doubling and seed subsequent rearrangement cycles [77]. Spatial transcriptomics co-maps clone-specific rearrangements with microenvironmental cues, revealing enrichment of chromoanagenesis-bearing cells at invasive fronts [78]. Key obstacles—allelic dropout, sparse coverage of copy-neutral variants, and heavy computational integration—still limit its routine application [79].

Regardless of the chosen method, comprehensive bioinformatic analyses are pivotal for reconstructing complex SVs and identifying their breakpoints [67,80]. Algorithms that integrate read depth, split-read mapping, and discordant paired-end reads help distinguish the hallmark clustered breakpoints associated with chromoanagenesis from more conventional stepwise mutations. By combining cytogenetic visualization with high-throughput sequencing and advanced computational approaches, researchers can gain a nuanced view of chromosomal catastrophes and their role in cancer biology [67].

## 4. Challenges and Future Directions

Despite refinements in chemotherapy since the 1980s, the overall survival rates of osteosarcoma have largely plateaued, highlighting the urgent need for new therapeutic strategies [1]. The increasing feasibility of molecular profiling, along with the future development of robust model systems and large, well-annotated tissue banks, will significantly advance our understanding of osteosarcoma biology. These forthcoming insights are expected to drive the development of innovative targeted therapies, focusing on tumor-specific molecular pathways and ubiquitously expressed surface antigens, with the potential to deliver more precise interventions and overcome the historically stagnant clinical outcomes [81].

The mechanistic understanding of chromoanagenesis needs to be further enhanced. While chromothripsis plays a critical role in the initiation and progression of osteosarcoma, the specific contributions of chromoanasynthesis and chromoplexy to its pathogenesis remain unclear. Therefore, large-scale international collaboration and single-cell multi-omics studies are needed to clarify how different types of chromoanagenic events (e.g., LTA chromothripsis) develop and evolve within osteosarcoma subtypes, as well as to quantify their functional impact on cell fitness [14]. Additionally, how CGRs influence cancer cell plasticity, immune evasion, and other tumor–microenvironment interactions remains unclear. Integrating emerging technologies such as CRISPR-based gene editing, real-time imaging, and high-throughput single-cell analysis could illuminate these processes in unprecedented detail.

However, the distinctive genomic complexity of osteosarcoma—dominated by extensive CNVs and SVs—poses a significant challenge to identifying reliable biomarkers and developing effective targeted therapies [6,44]. Although osteosarcomas exhibit significant genomic instability, including frequent chromothripsis and kataegis, they harbor few recurrent targetable mutations, and clinical trials with targeted agents have largely produced disappointing outcomes [82]. Emerging evidence points to a wealth of tumor-specific vulnerabilities embedded within CNVs and SVs that could be exploited for therapy. Realizing this potential has been hindered by limited preclinical models that accurately recapitulate the heterogeneity of osteosarcoma tumors [6]. Patient-derived tumor xenografts (PDTXs) offer a promising route to address this gap, as they retain key genetic and biological features of the primary tumors and can be used to test personalized, genome-informed treatment strategies [6,83,84,85]. Encouraging proof-of-concept studies have identified highly amplified oncogenes in patient samples, validated these amplifications in corresponding PDTX models, and demonstrated significant tumor responses using subclass-specific therapeutics [85]. In addition, bones harbor a distinct immune microenvironment governed by numerous signaling pathways essential for bone homeostasis. The success of the adjuvant innate immune stimulant mifamurtide in non-metastatic osteosarcoma strongly suggests that newer immune-based therapies, including immune checkpoint inhibitors, could markedly improve patient outcomes [82].

In the future, effective collaboration among bioinformaticians, molecular biologists, and clinicians will be instrumental in designing and implementing large-scale, multicenter studies to validate chromoanagenesis as both a biomarker and a therapeutic target. By merging advanced genomic tools, improved preclinical models, and innovative clinical trial designs, the osteosarcoma research community can work toward a new generation of therapies that disrupt the drivers of chromoanagenesis. Such an approach has the potential to break the decades-long therapeutic impasse and offer renewed hope for patients battling this challenging malignancy.

## 5. Conclusions

In summary, chromoanagenesis represents a pivotal mechanism driving the profound genomic instability characteristic of osteosarcoma. Catastrophic events such as chromothripsis, chromoplexy, and chromoanasynthesis—often facilitated by impaired DNA repair, replication stress, and mitotic errors—contribute to highly rearranged chromosomes that underlie tumor initiation, heterogeneity, and progression. Osteosarcoma is among the most striking examples of a malignancy driven by chromothripsis. Recent studies reveal that phenomena like “LTA chromothripsis” can trigger biallelic *TP53* inactivation while amplifying multiple oncogenes, thereby accelerating malignant transformation. Although osteosarcoma is notoriously complex and lacks broadly recurrent point mutations, the mounting evidence for widespread SVs highlights the significance of these large-scale rearrangements in shaping disease course and therapeutic responses.

Therefore, integrating high-depth genomic sequencing, single-cell multi-omics, and refined preclinical models promises to enhance our understanding of how these catastrophic genomic shifts evolve and drive osteosarcoma onset and progression. A deeper grasp of chromoanagenesis may pave the way for more targeted strategies, including personalized therapies directed against tumor-specific amplified oncogenes or pathways disrupted early in malignant transformation. By merging cutting-edge genomic tools with innovative clinical trial designs, the field is poised to move beyond the current therapeutic plateau and improve outcomes for patients facing this highly aggressive bone tumor.

## Figures and Tables

**Figure 1 biomolecules-15-00833-f001:**
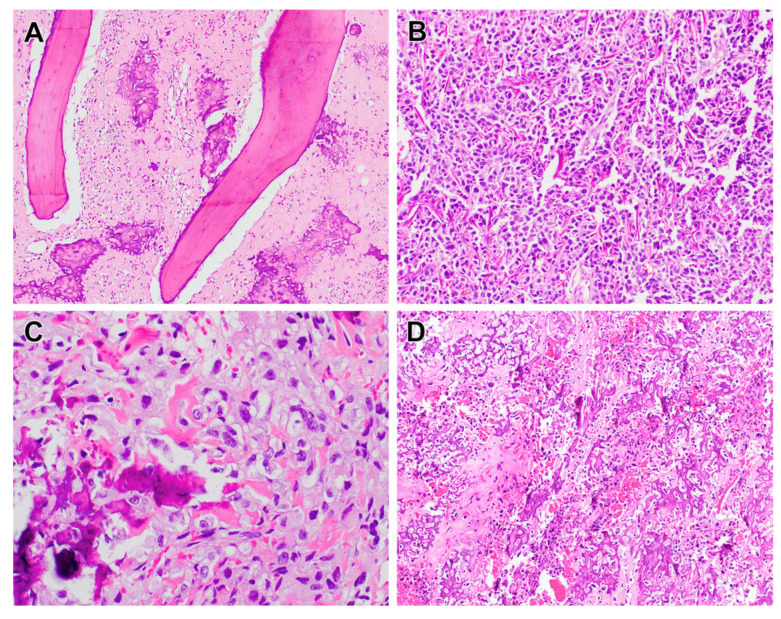
Conventional osteosarcoma is a high-grade bone-producing sarcoma which generally grows in an aggressive permeative fashion as seen here, (**A**) infiltrating through native cancellous bone trabeculae. The identification of mineralized osteoid is necessary for the diagnosis; mineralized osteoid may vary from wispy and lacelike (**B**) to coarse (**C**), abundant, and extensively sclerotic (**A**,**D**). (**A**) has a magnification of 40×, (**B**) has a magnification of 100×, (**C**) has a mag-nification of 200×, and (**D**) has a magnification of 100×.

**Figure 2 biomolecules-15-00833-f002:**
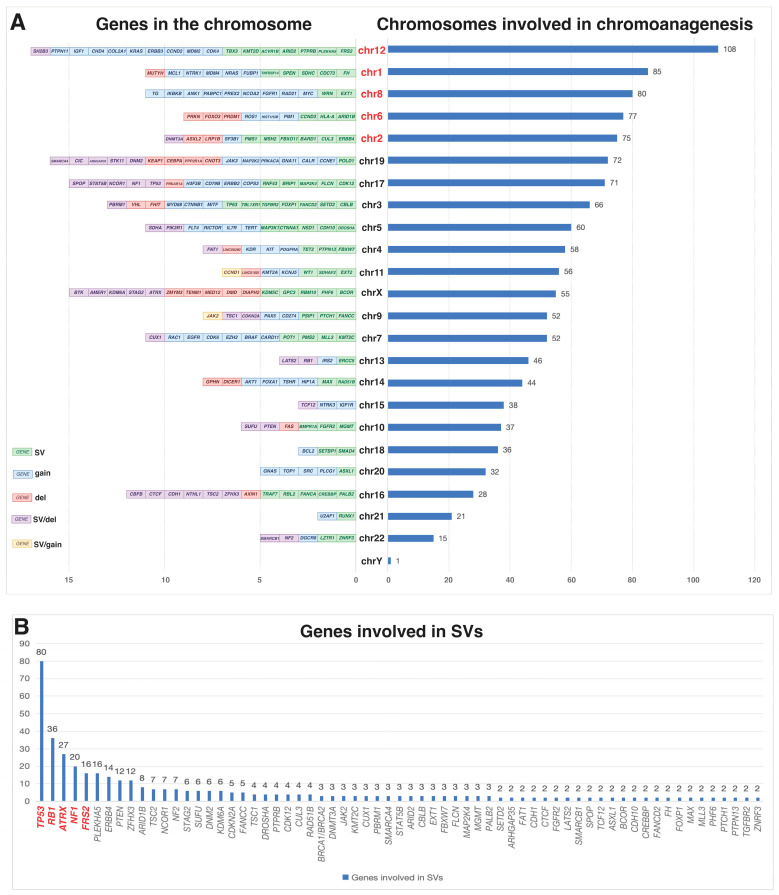
Chromosomes, genes, and variants in osteosarcoma with chromoanagenesis. (**A**) Genes in the chromosome (left side) and chromosomes (right side) involved in chromoanagenesis. (**B**) Genes with SVs. (**C**) Genes with CNVs. (**D**) Genes with SNVs.

**Figure 3 biomolecules-15-00833-f003:**
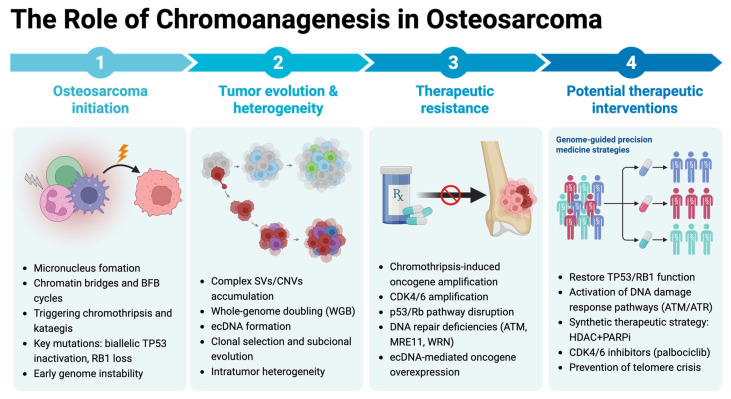
The role of chromoanagenesis in osteosarcoma. This conceptual model illustrates how chromoanagenesis initiates tumorigenesis, fuels clonal evolution and heterogeneity, underlies mechanisms of therapeutic resistance, and indicates potential therapeutic interventions. The figure was created using BioRender (http://biorender.com/, accessed on 22 May 2025).

**Figure 4 biomolecules-15-00833-f004:**
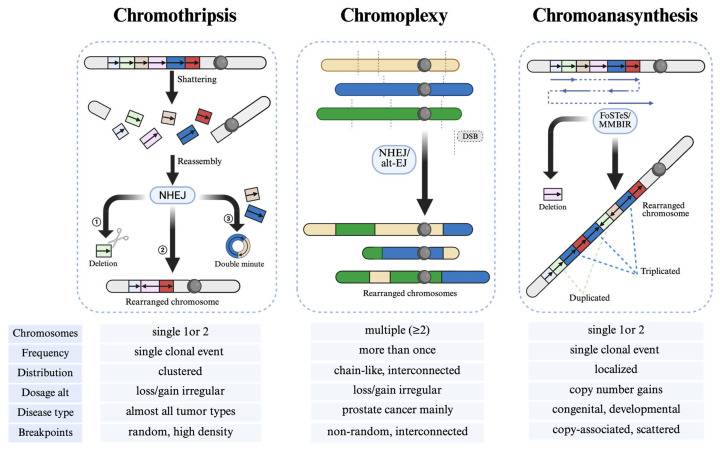
Mechanisms and differences among three types of chromoanagenesis. Chromoanagenesis comprises chromothripsis, chromoplexy, and chromoanasynthesis in terms of mechanisms and key features. Chromothripsis involves chromosomal shattering and reassembly via NHEJ. Chromoplexy leads to chain-like rearrangements across multiple chromosomes. Chromoanasynthesis results from replication errors, causing localized copy number gains. The table highlights their distinct characteristics. The figure was created using BioRender (http://biorender.com/, accessed on 17 February 2025).

**Figure 5 biomolecules-15-00833-f005:**
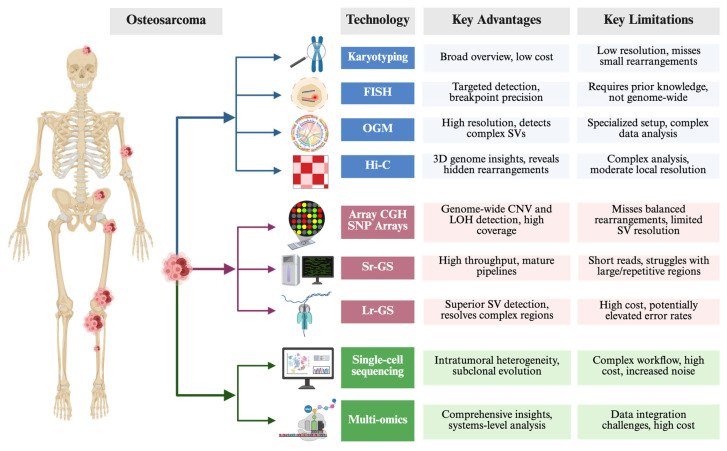
Technologies for detecting chromoanagenesis and their key advantages and limitations. Three categories of technologies for detecting chromoanagenesis are illustrated: cytogenetic methods (karyotyping, FISH, OGM, Hi-C), array-based and sequencing approaches (CGH/SNP arrays, short-read and long-read genome sequencing), and single-cell sequencing coupled with multi-omics profiling. Each method has distinct strengths, such as resolution, genome-wide coverage, or detection of SVs, as well as limitations, including cost, complexity, and detection biases. The table summarizes their key advantages and limitations. The tumors shown on the human skeleton indicate common sites of osteosarcoma occurrence, with larger tumor sizes being associated with a higher incidence. The figure was created using BioRender (http://biorender.com/, accessed on 8 March 2025).

## Data Availability

Not applicable.

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
