# Peer review of "Chromoanagenesis in Osteosarcoma"

_biomolecules, 2025, doi:10.3390/biom15060833_

Round 1
Reviewer 1 Report
Comments and Suggestions for Authors
In the present work, Li et al. reviewed the literature on the phenomenon of chromoanagenesis in osteosarcoma. The present work is well-written and I have enjoyed reading it. It is well-organized, and very interesting since it addresses a very important neoplasm, for which much is still unknown. The authors have included a very interesting section concerning evolutionary dynamics. It would be nice for the authors to elaborate a little bit more on this topic, especially on the various models that exist and can give possible explanation on the phenomena of chromoanagenesis, chromothripsis etc. Finally, I would suggest to the authors to discuss an interesting point; if the processes of chromo-re-organizing are random, how is it possible that after a chromosomal re-organization, a viable cellular offspring is produced? The process is probably random but the outcome is not random in the sense that an offspring compatible with life, means that chromosomes “still work”.
Overall, this is a very interesting and important review on the topic of osteosarcoma and its biology.
Reviewer 2 Report
Comments and Suggestions for Authors
The manuscript "Chromoanagenesis in Osteosarcoma" provides a comprehensive and well-structured review of catastrophic chromosomal rearrangements in osteosarcoma biology and therapeutic approaches. While the manuscript is well-organized, well-written, I have the following suggestions to enhance its overall impact and clarity:
- I highly recommend including a conceptual figure that visually depicts the role of chromoanagenesis in driving osteosarcoma initiation, tumor evolution, and mechanisms of therapeutic resistance.
- Figure 2 requires significant improvement in terms of readability - the current font size is too small to be useful. Professional reformatting would greatly enhance its value to readers.
- The prognostic implications of chromoanagenesis patterns deserve more thorough exploration. Although TP53 inactivation and LOH are discussed, please clarify whether specific chromoanagenesis signatures correlate with clinical outcomes or have predictive value.
- Expand the discussion on cutting-edge technologies, particularly single-cell sequencing and integrated multi-omics approaches. A table summarizing osteosarcoma-specific studies that utilize these methodologies would be a valuable addition to the review.
Reviewer 3 Report
Comments and Suggestions for Authors
This is an interesting review on the subject of chromoanagenesis in osteosarcoma. Overall, it covers most aspects of chromoanagenesis, particularly in the context of osteosarcoma, ranging from underlying mechanisms to clinical features. I would recommend its publication after some essential modifications:
- Different types of karyotype alterations—including chromothripsis, chromoplexy, chromoanasynthesis, giant nuclei (from PGCCs), and chromosome fragmentations—belong to the broader category of genome chaos. The general mechanism involves alterations to karyotype coding, a system-level form of inheritance that preserves the order of genes along and among chromosomes. The reorganization of the genome through genome chaos results in the formation of new genomes, marking the macroevolutionary phase of the two-phased model of cancer evolution. A brief discussion of this concept—the Genome Architecture Theory—would significantly enhance the impact of the review and help readers appreciate the importance of karyotype changes during the macroevolutionary phase of cancer.
- SKY (spectral karyotyping) or multicolor FISH are powerful tools for studying genome chaos and should be mentioned in this context (PMID: 30260497)
- It is known that doxorubicin treatment can effectively induce genome chaos. How does this apply to osteosarcoma treatment? Does such treatment contribute to rapid drug resistance via induced genome chaos (PMC8237085)?
